# CLP290 promotes the sedative effects of midazolam in neonatal rats in a KCC2-dependent manner: A laboratory study in rats

Akiko Doi[1], Tomoyuki Miyazaki[1,2]*, Takahiro Mihara[1], Maiko Ikeda[1], Ryo Niikura[1], Tomio Andoh[3], Takahisa Goto[1]

1 Department of Anesthesiology and Critical Care Medicine, Yokohama City University Graduate School of Medicine, Yokohama, Japan, 2 Department of Physiology, Yokohama City University Graduate School of Medicine, Yokohama, Japan, 3 Department of Anesthesiology, Mizonokuchi Hospital, Teikyo University School of Medicine, Kawasaki, Japan

* johney@yokohama-cu.ac.jp

**Data Availability Statement:** All relevant data are within the manuscript and its Supporting Information files.

## Abstract

Immature neurons dominantly express the $Na^+$-$K^+$-$2Cl^-$ cotransporter isoform 1 (NKCC1) rather than the $K^+$-$Cl^-$ cotransporter isoform 2 (KCC2). The intracellular chloride ion concentration ($[Cl^-]i$) is higher in immature neurons than in mature neurons; therefore, γ-aminobutyric acid type A (GABAA) receptor activation in immature neurons does not cause chloride ion influx and subsequent hyperpolarization. In our previous work, we found that midazolam, benzodiazepine receptor agonist, causes less sedation in neonatal rats compared to adult rats and that NKCC1 blockade by bumetanide enhances the midazolam-induced sedation in neonatal, but not in adult, rats. These results suggest that GABA receptor activation requires the predominance of KCC2 over NKCC1 to exert sedative effects. In this study, we focused on CLP290, a novel KCC2-selective activator, and found that midazolam administration at 20 mg/kg after oral CLP290 intake significantly prolonged the righting reflex latency even in neonatal rats at postnatal day 7. By contrast, CLP290 alone did not exert sedative effects. Immunohistochemistry showed that midazolam combined with CLP290 decreased the number of phosphorylated cAMP response element-binding protein-positive cells in the cerebral cortex, suggesting that CLP290 reverted the inhibitory effect of midazolam. Moreover, the sedative effect of combined CLP290 and midazolam treatment was inhibited by the administration of the KCC2-selective inhibitor VU0463271, suggesting indirectly that the sedation-promoting effect of CLP290 was mediated by KCC2 activation. To our knowledge, this study is the first report showing the sedation-promoting effect of CLP290 in neonates and providing behavioral and histological evidence that CLP290 reverted the sedative effect of GABAergic drugs through the activation of KCC2. Our data suggest that the clinical application of CLP290 may provide a breakthrough in terms of midazolam-resistant sedation.

**Funding:** This project was supported by Japan Society for the Promotion of Science, Grant-in-Aid for Scientific Research (C)(16K10945) to TA. The funders had no role in study design, data collection and analysis, decision to publish, or preparation of the manuscript.

**Competing interests:** The authors have declared that no competing interests exist

## Introduction

γ-Aminobutyric acid (GABA) is an inhibitory neurotransmitter that binds to the GABA receptor. In most mature neurons, GABAA receptor activation (e.g., by anesthetic agents) opens a $Cl^-$ channel and drives chloride ion influx in an electrochemical gradient-dependent manner, which subsequently leads to neuronal hyperpolarization [1–3]. By contrast, immature neurons maintain relatively higher intracellular chloride ion levels ($[Cl^-]i$) compared to those in mature neurons and, therefore, GABAA receptor activation does not induce chloride ion influx [4,5]. Instead, it may induce chloride ion efflux, leading to neuronal depolarization [1,6]. The underlying mechanism regulating $[Cl^-]i$ involves the balance of $Na^+$-$K^+$-$2Cl^-$ cotransporter isoform 1 (NKCC1) and $K^+$-$Cl^-$ cotransporter isoform 2 (KCC2) levels [7]. In the developing rodent brain, especially the cerebral cortex and hippocampus, NKCC1 is more dominantly expressed and induces $Cl^-$ entry into the cytoplasm compared to KCC2 which mainly mediates $Cl^-$ excretion. Therefore, compared to mature neurons, immature neurons in these regions present relatively higher $[Cl^-]i$. In the neonatal rat cerebral cortex, the NKCC1 expression level is usually highest around postnatal day (PND) 5 to 7; afterward, the KCC2 expression level gradually increases and becomes dominant [1]. The switch in the functional role of the GABAA receptor from excitatory to inhibitory GABA signaling occurs between PND8 and 14, which correlates with the switch in dominant expression from NKCC1 to KCC2 [6–8].

Previous studies have reported that midazolam, an allosteric agonist of the GABAA receptor that binds to the benzodiazepine receptor, does not exert a sedative effect in PND3 rats at similar doses that induce hypnosis in PND21 rats [9]. Similarly, phenobarbital, another allosteric modulator of the GABAA receptor, has been reported to have a significant anticonvulsive effect in adult rodents but not in neonatal rats at PND6 to 12 [1]. These findings indicate that GABAA receptor activation does not generally induce neuronal inhibition in neonatal animals [10]. Notably, pretreatment with bumetanide, an NKCC1 inhibitor, has been reported to revert the sedative effect of midazolam and the anticonvulsant effect of phenobarbital even in neonatal rats [6,11,12]. These findings indicate that NKCC1 inhibition can promote the inhibitory activity of GABAA receptor agonists. However, it remains unclear whether KCC2 activation exerts the same effect as NKCC1 inhibition in the neonatal period.

NKCC1 and $K^+$-$Cl^-$ cotransporter isoforms other than KCC2 are expressed in the brain, as well as in several other organs, e.g., the kidney and blood vessels [13]. The clinical use of compounds acting on NKCCs and KCC3/KCC4 was reported to induce adverse effects, including diuresis or hypokalemic alkalosis, resulting in multiple organ dysfunctions. By contrast, KCC2 is expressed exclusively in the central nervous system [3,14]; therefore, $[Cl^-]i$ manipulation through the KCC2 pathway could minimize adverse effects [15]. Consequently, KCC2 has been proposed as a feasible therapeutic target for neuronal disorders, including neuropathic pain, epileptic seizure, neurological trauma, and metabolic diseases, in which KCC2 dysfunction might be the underlying cause [13,16,17]. Gagnon et al. successfully developed two KCC2-selective analogs: CLP257 and CLP290 which is a carbamate prodrug of CLP257. CLP257 increases the cell surface expression of KCC2 and decreases $[Cl^-]i$ in cultured injured neurons. Furthermore, in vivo experiments showed that systemic CLP290 administration in a rat model of neuropathic pain can improve the hyperalgesia threshold [13]. However, to our knowledge, the effects of these compounds on immature neurons are still uncertain, and no study has shown the sedation-promoting effects of these compounds in neonatal animals.

We hypothesized that activation of KCC2 promotes the sedative effect of GABAA receptor activation in immature rodents. To test this hypothesis, we assessed the righting reflex behavior of neonatal and adult animals treated with midazolam and CLP290. Next, we conducted phosphorylated cyclic adenosine monophosphate-response element-binding protein (p-

CREB) immunohistochemistry to determine the brain region related to the CLP290 effect on midazolam-induced sedation. Moreover, to elucidate the underlying mechanism, we investigated whether the sedation-promoting effect of CLP290 in neonates can be antagonized by the KCC2-selective antagonist VU0463271. Finally, we measured the cell surface expression of KCC2 to elucidate the biochemical processes underlying the in vivo effect of CLP290.

## Materials and methods

### Animals

This study was carried out in strict accordance with the recommendations in the Guide for the Care and Use of Laboratory Animals of the Yokohama City University, created based on the Guidelines for Proper Conduct of Animal Experiments of the Science Council of Japan. The protocol of this study was approved by the Committee on the Ethics of Animal Experiments of the Yokohama City University (Study protocol Number: F-A-16-045). We used male Sprague-Dawley rats (Japan SLC, Shizuoka, Japan); specifically, PND7 neonatal rats ($n = 161$) and 4-week-old adult rats ($n = 70$). We kept neonatal rats in cages with their littermates and mothers; the adult rats were housed in groups of two or three. The rats were housed in plastic cages placed in a temperature- and humidity-controlled animal care facility room with a 14-h/10-h light/dark cycle (light on from 5:00 to 19:00). The rats were allowed ad libitum access to food and water. The welfare of all rats was checked daily at the time of feeding, and we made all efforts to minimize animal suffering and the number of animals used.

### Drugs

On the day of the experiment, midazolam (Sandoz, Tokyo, Japan) was diluted with 5× saline (final 1 mg/ml) to be administered at 0.02 ml/g of body weight. CLP290 was received from Dr. Yves De Koninck (Institut Universitaire en Santé Mentale de Québec) or purchased (#SML-2172, Sigma-Aldrich, St. Louis, MO, USA). CLP290 was adjusted to a final concentration of 0.66% prior to oral administration; the CLP290 from Dr. De Koninck was dissolved in 10% hydroxypropyl-β-cyclodextrin, whereas the CLP290 from Sigma-Aldrich was suspended in 0.5% carboxy methylcellulose. On the day of the experiment, VU0463271 (#4719/10, Tocris Bioscience, Bristol, UK) was dissolved in 0.75% dimethyl sulfoxide, at a final concentration of 1 mM.

### Behavioral procedures

A righting reflex test was conducted to assess in neonatal and adult rats their sedation level at various doses and combinations of experimental drugs [18]. This experiment was performed in a similar manner, as previously described [12,19]; in short, rats were gently placed on their backs, and the righting reflex was measured based on the latency to attain the upright position. The predetermined latency time threshold was 60 s, and the rats were returned to the upright position each time. The measurements were obtained in triplicate, and the mean latency was calculated to determine the righting reflex latency. The following three experiments were conducted:

1. Thirty minutes prior to the righting reflex test, neonatal rats were randomly assigned to seven groups, followed by intraperitoneal administration of midazolam at a dose of 0, 5, 10, 20, 30, 50, or 70 mg/kg ($n = 9$). Similarly, adult rats were assigned to six groups, followed by intraperitoneal administration of midazolam at a dose of 0, 5, 10, 20, 30, or 40 mg/kg ($n = 7$).

2. To assess the promotion effect of CLP290 on midazolam-induced sedation in each developmental stage, neonatal and adult rats were randomly assigned to two groups, and CLP290 (100 mg/kg) or an equal volume of vehicle was orally administered. Subsequently, 2 h after CLP290 or vehicle administration, each group was subdivided into two groups. Neonate rats were intraperitoneally injected with midazolam at 20 mg/kg or an equal volume of saline, as previously reported [12] ($n = 5$), and adult rats received intraperitoneally midazolam at 8.6 mg/kg, which is equivalent to a 20% effective dose ($ED_{20}$) obtained from Fig 1D, or an equal volume of saline ($n = 7$). The righting reflex latency of all rats was measured 30 min after administration of midazolam or saline.

3. To assess the involvement of KCC2 in the promotion effect of CLP290 on midazolam-induced sedation in neonatal rats, rats were pretreated with CLP290 followed by VU0463271 prior to the midazolam administration. That is, neonatal rats were randomly assigned to two groups and orally pretreated with CLP290 (100 mg/kg) or an equal volume of vehicle 2 h prior to the midazolam administration. Subsequently, each group was subdivided into two groups, and VU0463271 or an equal volume of vehicle was injected twice intraperitoneally at 0.02 ml/g and 0.01 ml/g 20 min and 50 min, respectively, after the first pretreatment. All rats received midazolam at 20 mg/kg intraperitoneally 70 min after the second administration of VU0463271 or vehicle. The righting reflex latency of all rats was measured 30 min after administration of midazolam ($n = 6$).

## Immunohistochemistry for p-CREB detection

This immunohistochemistry experiment was conducted to assess the neuronal activity following the separate or combined administration of midazolam and CLP290. Neonatal rats were randomly assigned to two groups and orally treated with CLP290 (100 mg/kg) or an equal volume of vehicle. Subsequently, 2 h after CLP290 or vehicle administration, each group was subdivided into two groups to receive midazolam at 20 mg/kg or an equal volume of saline intraperitoneally. At 45 min after midazolam or saline injection, the rats were deeply anesthetized with an intraperitoneal injection of sodium pentobarbital at 324 mg/kg (Somnopentil, 64.8 mg/ml, Kyoritsuseiyaku, Tokyo, Japan) and immediately perfused and fixed, as previously described [12]. In short, rats were perfused transcardially with 5 ml/10 g body weight of 0.1 M phosphate-buffered saline (PBS) for 5 min, followed by 10 ml/10 g body weight of 4% paraformaldehyde-PBS pH 7.4 for 10 min. Subsequently, the animals were decapitated, and their brains were postfixed in 4% paraformaldehyde-PBS at 4˚C for 2 h and cryoprotected in 20% sucrose. Coronal brain sections of 20 μm thickness were obtained at the hippocampal level using an 1800 Cryostat Microtome (Leica Microsystems, Wetzlar, Germany). Prior to staining, slices were treated with 0.3% $H_2O_2$ in methanol to suppress endogenous peroxidase activity. After blocking in tris-buffered saline (TBS) containing 2% normal donkey serum and 0.5% Triton X-100 for 1 h at room temperature, the slices were incubated overnight at 4˚C with primary polyclonal rabbit anti-p-CREB (1:500; #06–519; Millipore, Burlington, MA, USA). After washing the slices three times, they were incubated for 1 h at room temperature with biotinylated anti-rabbit secondary antibody (1:200; #711-066-152; Jackson Immuno Research, West Grove, PA, USA). After washing the slices again three times, they were incubated with avidin-biotin-horseradish peroxidase solution (Vectastain ABC Elite kit, Vector Laboratories, Burlingame, CA, US) for 1 h and developed using 3,3'-diaminobenzidine. Then, an operator blinded to the treatments performed the measurements using a Biorevo BZ 9000® microscope (Keyence Corporation, Osaka, Japan) as follows. First, three hippocampal-level slices were collected from each brain according to the region of Fig 36 of the Paxinos and Watson atlas [20]. The

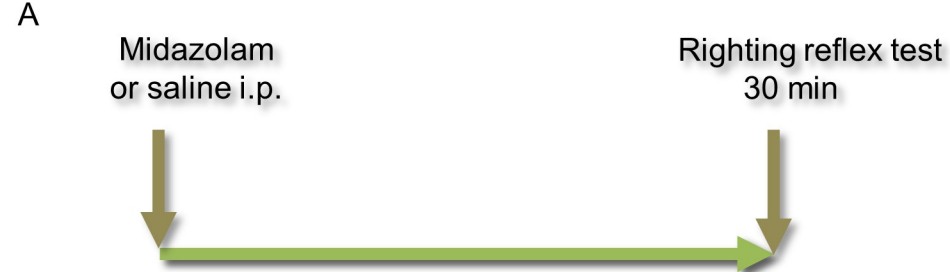

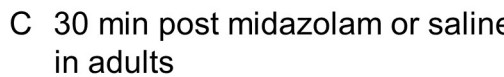

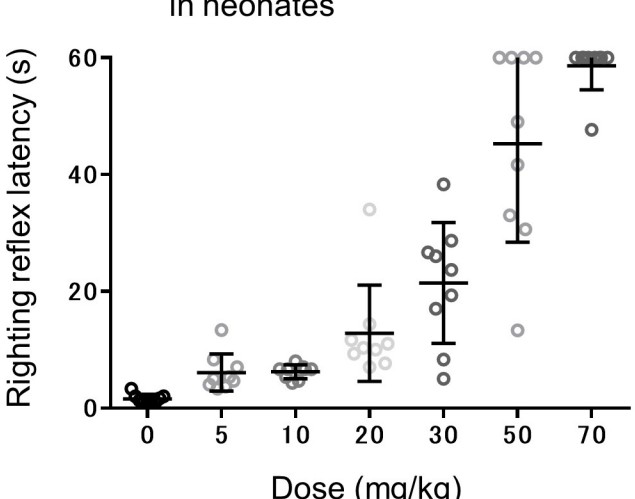

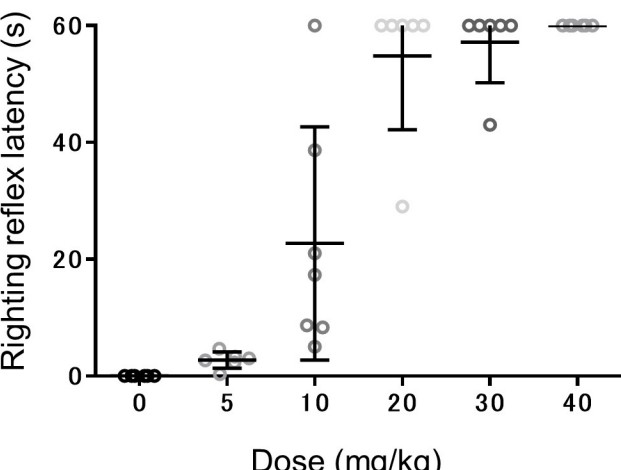

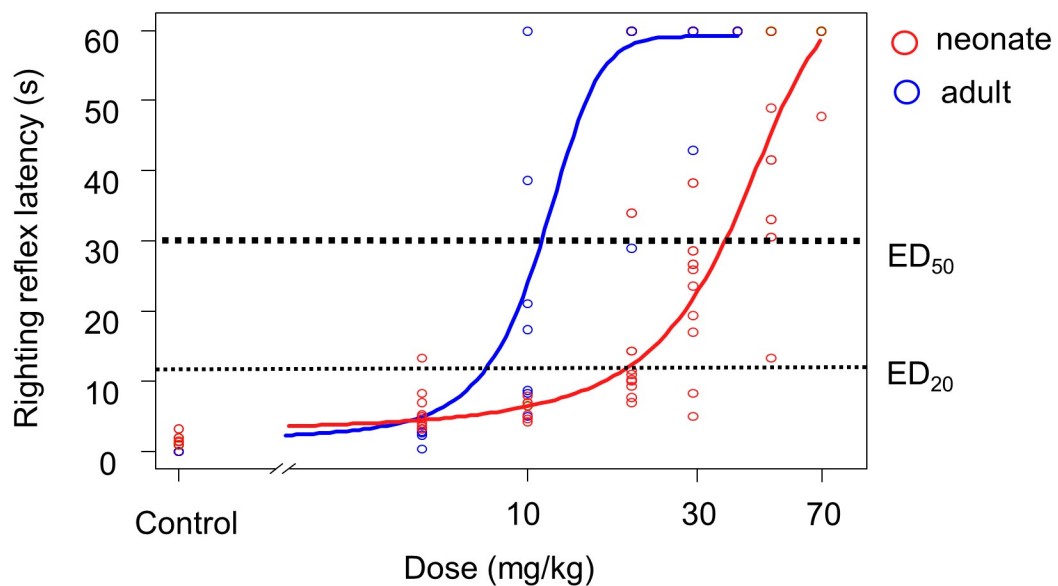

**Fig 1. Neonatal rats require higher doses of midazolam for sedation than adult animals.** (A) Time schedule of drug administration and behavioral test. (B) Righting reflex latency at 30 min after administration of midazolam at different doses in neonates ($n = 9$). (C) Righting reflex latency at 30 min after

administration of midazolam at various doses in adult rats ($n$ = 7). Transverse and vertical bars indicate the mean and standard deviation [1], respectively; dots indicate individual values. (D) Dose-response curve of the righting reflex latency at 30 min after administration of midazolam in neonate and adult animals. Dotted lines indicate $ED_{50}$ and $ED_{20}$ values. $ED_{50}$: adults 11.03 mg/kg; neonates, 38.58 mg/kg; $p < 0.001$. $ED_{20}$: adults, 8.6 mg/kg; neonates, 20.0 mg/kg.

somatosensory cortex and thalamus regions were imaged from each slice. The p-CREB-positive cells at $100 \times 200$ μm$^2$ in the S1 barrel field and $200 \times 500$ μm$^2$ in the ventral thalamus were counted. The cell numbers in each region in three slices were determined, and the average was calculated.

## Biochemical analysis

Biochemical analyses were conducted to assess the CLP290 effect on KCC2 expression. Neonatal rats were orally administered 100 mg/kg of CLP290 or an equal volume of vehicle. After 30 min and 60 min, which corresponds to the highest plasma level [13], rats were deeply anesthetized with isoflurane, followed by decapitation performed by trained personnel. The brains were quickly collected and transferred into ice-cold dissection buffer (0.25 mM NaH$_2$PO$_4$□2 H$_2$O, 0.5 mM KCl, 0.1 mM CaCl$_2$, 1.4 mM MgCl$_2$□6 H$_2$O, 18 mM choline chloride, 11.6 mM ascorbic acid, 3.1 mM pyruvic acid, 25 mM NaHCO$_3$, 25 mM glucose, gassed with 5% CO$_2$/ 95% O$_2$). Using a brain slicer (Brain Matrix, #G8105, EM Japan, Tokyo, Japan), 1 mm thick slices were prepared at the level of the rat brain atlas Figs 42–49 [21], i.e., 1.6–3.0 mm behind the bregma. Referring to this rat brain atlas, the S1 barrel field was cut out as the somatosensory cortex. Subsequently, the slices were incubated with artificial cerebrospinal fluid (0.5 mM KCl, 23.6 mM NaCl, 0.2 mM NaH$_2$PO$_4$□2 H$_2$O, 25 mM NaHCO$_3$, 25 mM glucose, 2 mM MgCl$_2$, 2 mM CaCl$_2$) containing 2 mg/ml Sulfo-NHS-Biotin (Thermo Fisher Scientific, Waltham, MA, USA) at 4°C for 45 min with slow rotation. After rinsing three times with TBS, slices were homogenized with immunoprecipitation buffer (150 mM NaCl, 0.5 mM EDTA, 0.1 mM EGTA, 1 mM HEPES, 1% Triton-X), and the homogenate was centrifuged at $14000 \times g$ for 15 min at 4°C. The protein concentration in the supernatant was measured using the Pierce bicinchoninic acid protein assay kit (Thermo Fisher Scientific) and adjusted to a similar protein concentration for all samples. To determine the total fraction, 30 μl aliquots of the solution were mixed with an equal amount of 2× sample buffer and denatured at 100°C for 5 min. To determine the membrane fraction, a 150 μl aliquot of the adjusted solution was mixed with an equal volume of NeutrAvidin agarose resin (Thermo Fisher Scientific). The mixed solutions were then incubated at 4°C for 16 h with rotation followed by centrifugation at $2000 \times g$ for 1 min at 4°C. Subsequently, the supernatant was removed, the resulting gel was washed five times, followed by mixing with 150 μl 2× sample buffer and denaturation at 100°C for 5 min. A similar amount of protein (2 μg total protein and 4 μg membrane protein) was subjected to 7.5% SDS-PAGE followed by transfer to polyvinylidene fluoride membrane. The membranes were blocked for 2 h with 2% ECL blocking agent (#RPN418, GE Healthcare, Chicago, IL, USA) in TBS-T and then incubated overnight at 4°C with anti-KCC2 (1:5000; #07–432, Millipore) and anti-β-actin (1:10000; #A5441, Sigma-Aldrich) antibodies. The blocking agent was used to prepare all antibody reagents. After washing, membranes were incubated for 1 h with secondary antibody (1:10000; #175–6515, 170–6520, BioRad, Hercules, CA, USA). Protein visualization was conducted using the ECL detection system (#RPN2235: 2236, GE Healthcare).

## Statistical analyses

GraphPad prism 6 and R version 3.4.3 (The R Project for Statistical Computing, November 2017; www.r-project.org) was used for statistical analysis and figure generation.

The midazolam dose-response curves in neonate and adult rats were analyzed using a three-parameter logistic function. From the dose-response curves, the 50% effective dose ($ED_{50}$: indicating the dose at which the righting reflex latency is 30 s) was calculated separately for neonates and adults. The difference in $ED_{50}$ between the two groups was analyzed using the Z-test.

The results of the behavioral test are presented as the median values of individual groups. The median regression model included two treatment factors, and their interaction was analyzed for each experiment (midazolam and CLP290 in Figs 2 and 3, CLP290 and VU0463271 in Fig 5). Upon significant interaction, the differences between the baseline group and the other groups were assessed using median regression analysis with the midazolam + CLP290 group in Figs 2 and 3 and the CLP290 + VU0463271 + midazolam group in Fig 5 defined as the baseline groups.

The numbers of p-CREB-positive cells in the neonatal rats are presented as the mean values for individual groups and were analyzed using the Tukey-Kramer test.

In the biochemical analyses, the protein expression levels in the crude fractions were normalized to the β-actin level, and the expression in the membrane fraction was determined as the ratio to that in the crude fraction. Furthermore, individual values were normalized to the mean control values (presented as 1). The Mann-Whitney U test was used for subsequent analyses.

We defined a significant difference at $p < 0.05$.

## Results

### CLP290 promotes the sedative effect of midazolam in neonatal rats

To elucidate the difference in sedative midazolam effects on neonatal and adult rats, the dose-response relationship was studied 30 min after midazolam administration using the righting reflex latency as the parameter (Fig 1A). The $ED_{50}$ in neonatal rats was 38.58 mg/kg, whereas the $ED_{50}$ in adult rats was 11.03 mg/kg (Fig 1B and 1C). The Z-test revealed that the $ED_{50}$ in neonates was significantly higher than that in adult animals (Fig 1D). This result indicates that higher doses of midazolam are required for sedation in neonates compared to adult rats.

Next, to examine whether KCC2 activation promotes the sedative effect of midazolam in neonates and adults, CLP290 was administered, a KCC2 activator already used in several publications [10,13,15–17,22]. To better understand the difference in the sedation-promoting effect of CLP290 and that of bumetanide previously shown by Koyama et al., we considered midazolam at 20 mg/kg was suitable for the subsequent assay in neonates [12,19]. Based on the results shown in Fig 1D, we decided that in the righting reflex test, $ED_{20}$ concentrations were suitable for the comparison of CLP290 effects between neonates (midazolam 20 mg/kg) and adults (midazolam 8.6 mg/kg). In PND7 rats, showing resistance to midazolam-induced sedation, CLP290 pre-administration significantly prolonged the righting reflex latency compared to vehicle at 30 min after midazolam injection (Fig 2B). Moreover, the administration of CLP290 alone had no effect on the righting reflex latency (Fig 2B). This suggests that prior activation of KCC2 by CLP290 might promote the sedative effect of midazolam in neonates whereas CLP290 has no sedative effect by itself. Furthermore, the sedation-promoting effect of CLP290 in adult rats subjected to midazolam was examined (Fig 3A). In contrast to neonates, there was no interaction between midazolam and CLP290 regarding the righting reflex latency 30 min after midazolam treatment, suggesting that prior administration of CLP290 exerted little influence on the sedative effect of midazolam in adult rats (Fig 3B).

A

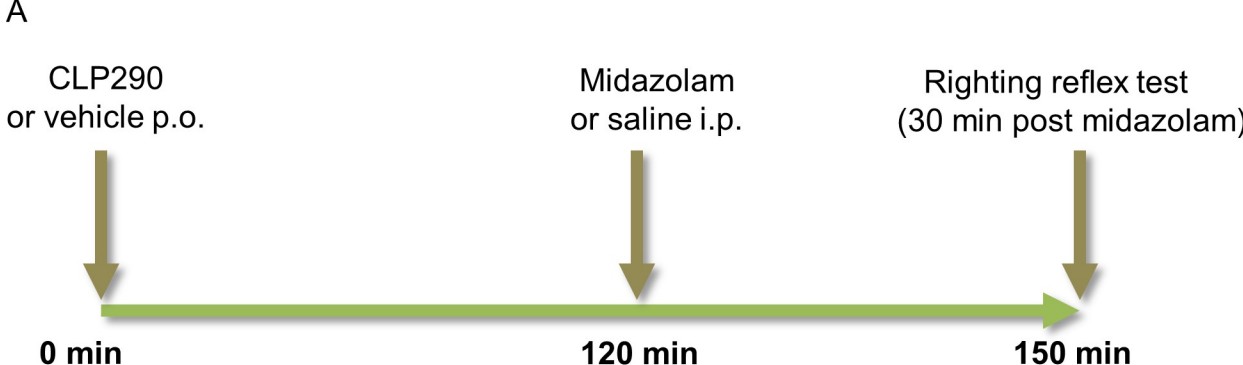

B 30 min post midazolam or saline in neonates

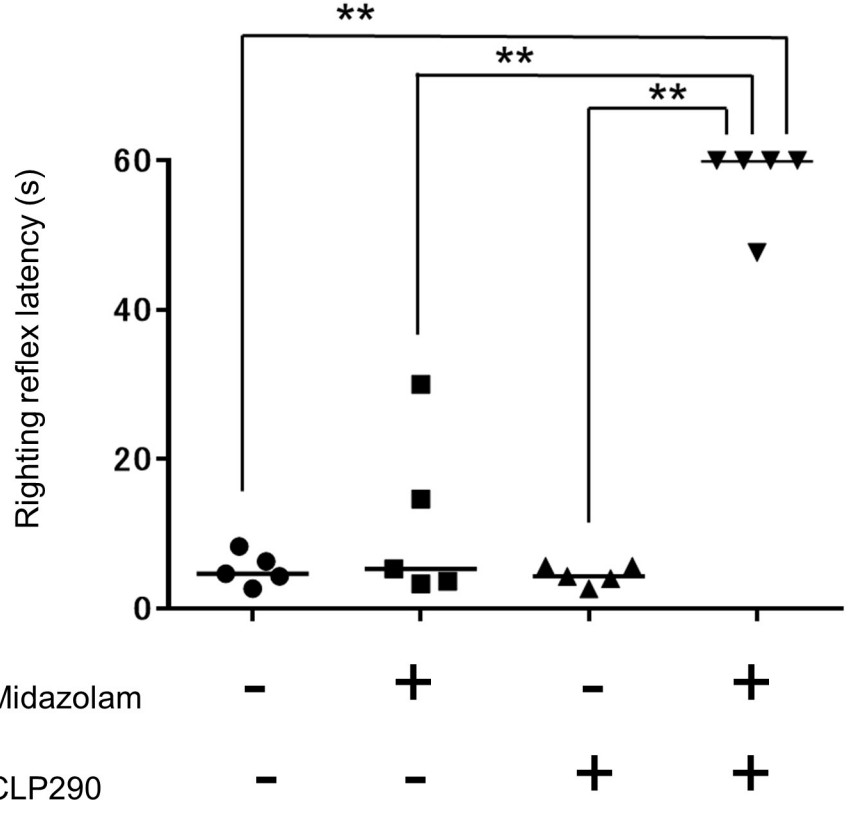

**Fig 2. CLP290 promotes the sedative effects of midazolam in neonatal rats.** Comparison of the righting reflex latency among four combinations using two drugs. (A) Time schedule of drug administration and behavioral test. (B) Righting reflex latency at 30 min after administration of midazolam or saline. The median latency of each group was as follows: vehicle + saline 4.7 s, vehicle + midazolam 5.3 s, CLP290 + saline 4.3 s, CLP290 + midazolam 60.0 s; ** indicates $p < 0.001$ (median regression analyses), $n$ = 5 animals/group. Transverse bars indicate median values; dots indicate individual values.

A

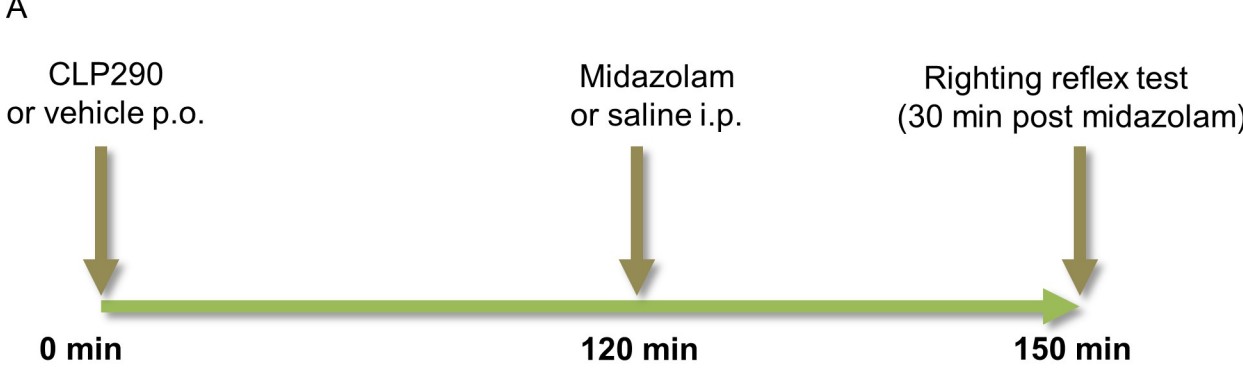

CLP290
or vehicle p.o.

Midazolam
or saline i.p.

Righting reflex test
(30 min post midazolam)

0 min          120 min          150 min

B  30 min post midazolam or saline in adults

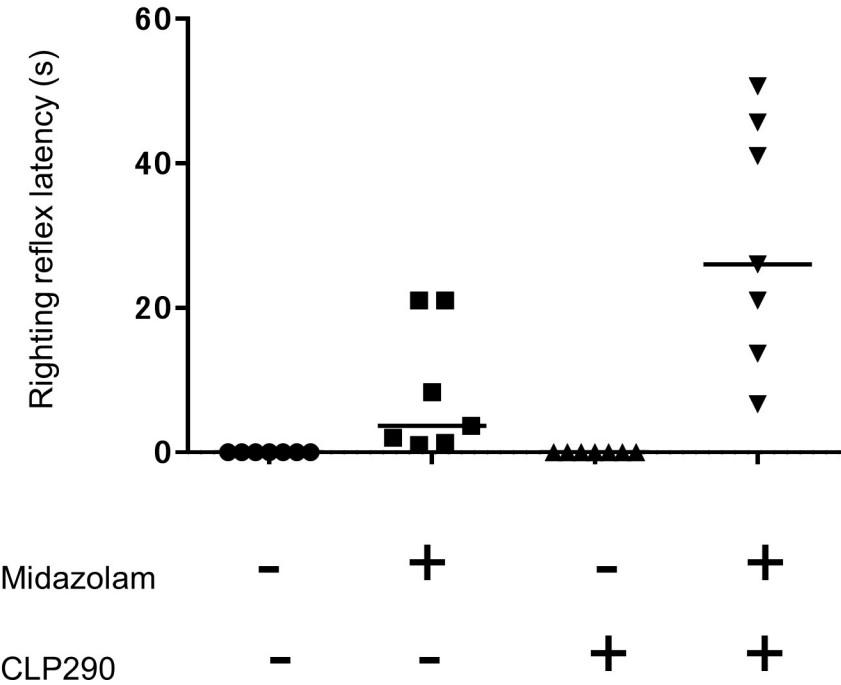

**Fig 3. Influence of CLP290 on the sedative effects of midazolam in adult rats.** Comparison of the righting reflex latency among the four drug combinations. (A) Time schedule of drug administration and behavioral test. (B) Righting reflex latency at 30 min after administration of midazolam or saline. Median latencies: vehicle + saline 0.00 s, vehicle + midazolam 3.67 s, CLP290 + saline 0.00 s, midazolam + CLP290 26.00 s, $n$ = 7 animals/group. Transverse bars indicate median values; dots indicate individual values.

## Coadministration of CLP290 with midazolam downregulates p-CREB expression in the neonatal somatosensory cortex

To identify brain regions involved in the CLP290 effect on midazolam-induced sedation, p-CREB immunohistochemistry was conducted (Fig 4A). There was no significant difference in the number of p-CREB-positive cells in the somatosensory cortex of neonatal rats after vehicle + midazolam and vehicle + saline administration (Fig 4B, 4C and 4F). By contrast, midazolam administration to neonatal rats pretreated with CLP290 significantly decreased the number of p-CREB-positive cells compared with vehicle administration (Fig 4C, 4E and 4F). Moreover, the administration of CLP290 alone had no effect on the number of p-CREB-positive cells (Fig 4B, 4D and 4F). These findings are inversely correlated to the level of sedation as assessed by the loss of the righting reflex (Fig 2). However, the numbers of p-CREB-positive cells in the thalamus were not significantly different among groups (S1 Fig). These findings showed that a single dose of 20 mg/kg midazolam had little effect on neuronal activity in the cerebral cortex and thalamus of PND7 rats. Midazolam administration after pretreatment with CLP290 exerted inhibitory effects on neuronal activity exclusively in the somatosensory cortex of neo-natal rats, which is consistent with the behavioral results in the righting reflex test shown in Fig 2.

## The CLP290 effect on midazolam-induced sedation is KCC2-dependent

To examine whether CLP290 promoted the sedative action of midazolam through KCC2 activation, we assessed the effect of VU0463271, a KCC2-selective inhibitor, on CLP290-pretreated neonatal rats. We determined the drug administration protocol based on the following reasons:

1. VU0463271 was administered intraperitoneally, whereas CLP290 was administered orally. Since it is expected that the increase in blood concentration will be slower with oral administration than with intraperitoneal administration, the concentration of CLP290 in the brain was expected to increase slower than that of VU0463271, as well as in the blood.

2. The in vivo half-life time of VU0463271 is 9 min [23]; therefore, repeated VU0463271 administration was required for the complete inhibition of the CLP290 effect.

Neonatal rats received CLP290 or vehicle followed by two times of VU0463271 or vehicle treatment. Finally, all rats were injected with midazolam prior to the righting reflex test (Fig 5A). There was a significant interaction between the effect of CLP290 and VU0463271 ($p = 0.011$). The main effect of CLP290 was significant ($p < 0.001$), whereas that of VU0463271 was not significant ($p = 0.785$). A post-hoc median regression analysis was performed as follows: First, the righting reflex latency after midazolam injection with CLP290 pretreatment was significantly longer than that without CLP290 administration (Fig 5B). Furthermore, VU0463271 coadministration significantly attenuated the effect of CLP290 on the righting reflex (Fig 5B). These results suggest that the effect of CLP290 on midazolam-induced sedation in neonatal rats is largely KCC2-dependent.

## CLP290 does not alter the expression and intracellular distribution of KCC2 in the neonatal somatosensory cortex

A previous study reported that CLP257 increases the cell surface expression of KCC2 in injured spinal neurons with subsequently decreased [Cl⁻]i [13]. To examine whether CLP290 altered the KCC2 expression pattern in the neonatal brain, we measured KCC2 levels in the crude and membrane fractions obtained from neonatal rats with or without CLP290

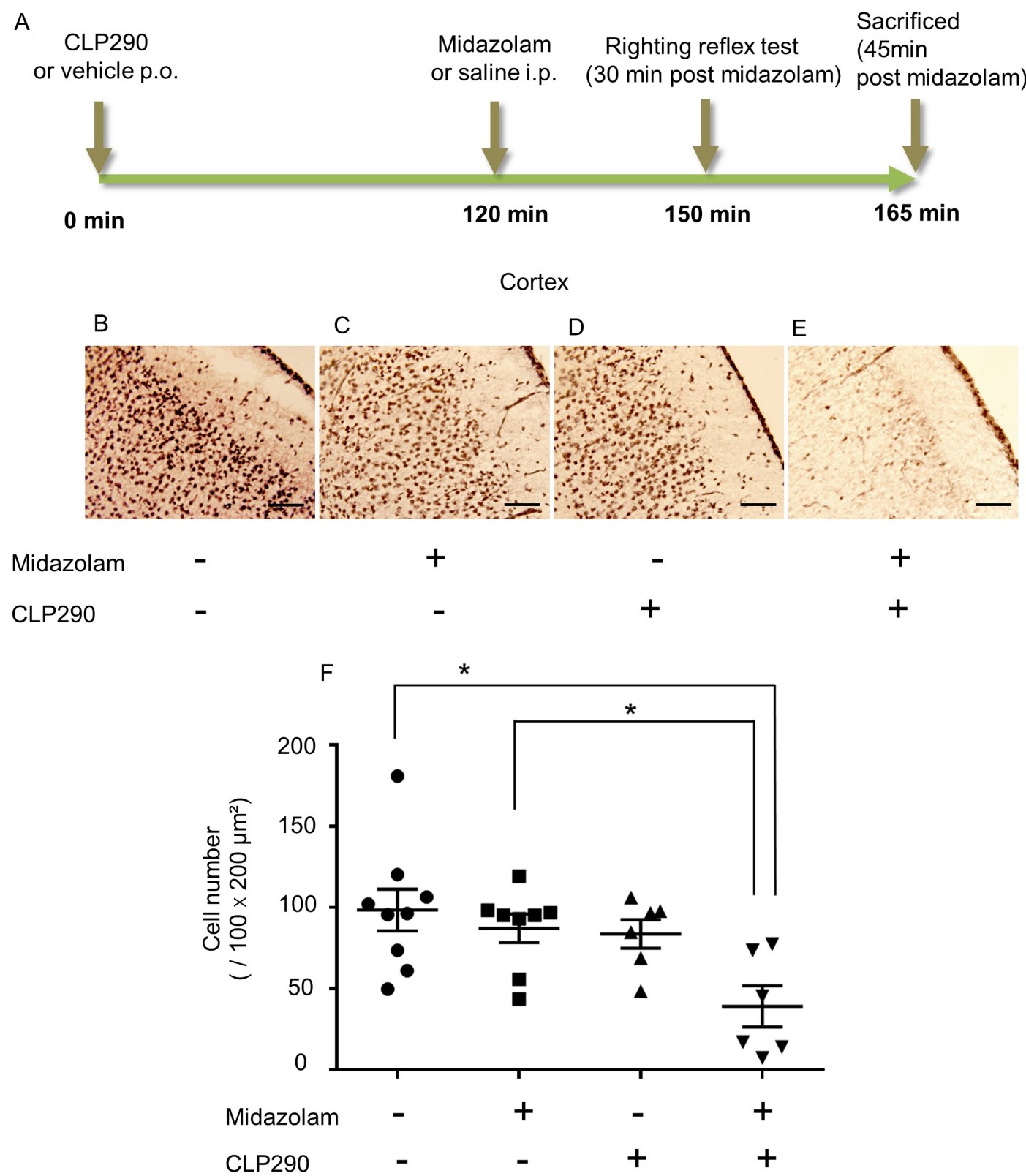

**Fig 4. Coadministration of midazolam with CLP290 downregulates p-CREB expression in the neonatal somatosensory cortex.** (A) Time schedule of drug administration, behavioral test, and immunohistochemistry. (B-E) Representative pictures of p-CREB expression in the neonatal somatosensory cortex for the following groups: (B) vehicle + saline, (C) vehicle + midazolam, (D) CLP290 + midazolam, (E) CLP290 + midazolam. (F) Mean numbers of p-CREB-positive cells in the somatosensory cortex for the four experimental groups: vehicle + saline 98.42 ($n = 9$), vehicle + midazolam 87.15 ($n = 8$), CLP290 + midazolam 83.67 ($n = 6$), CLP290 + midazolam 39.14 ($n = 6$). * indicates $p < 0.05$ (Tukey-Kramer test). Transverse and vertical bars indicate the mean and standard error (SE), respectively; dots indicate individual values.

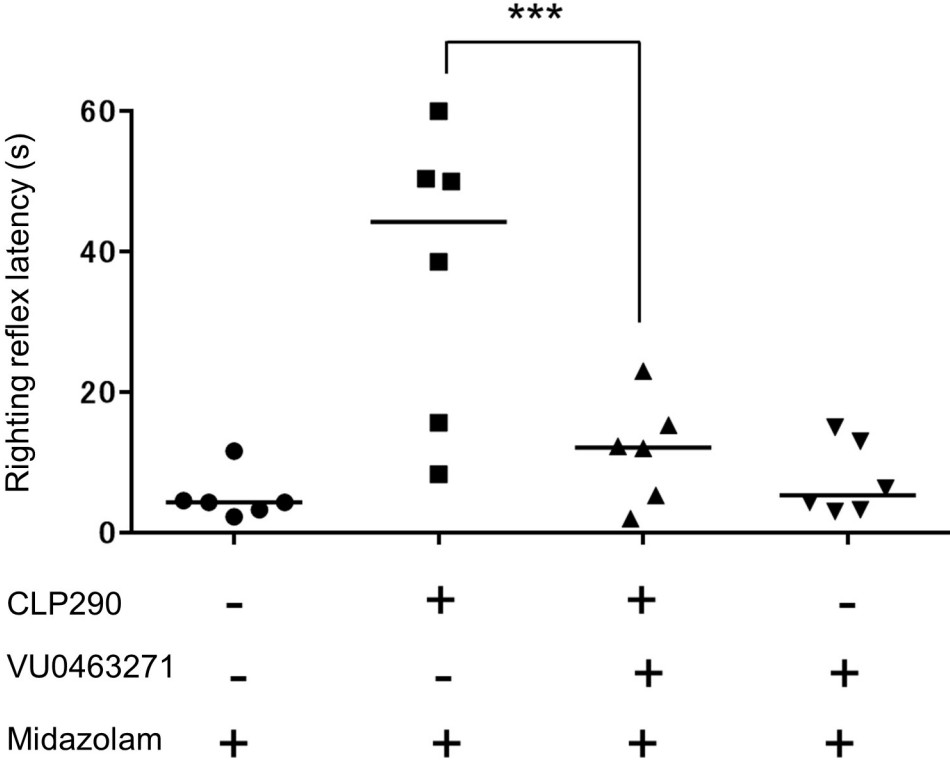

**Fig 5. CLP290 effect on midazolam-induced sedation is KCC2-dependent.** (A) Time schedule of drug administration and behavioral test. (B) The effect of multiple drug combinations on the righting reflex latency 30 min after midazolam administration. Median latencies of each group: vehicle + midazolam 4.3 s, CLP290 + midazolam 44.3 s, CLP290 + VU0463271 + midazolam 12.2 s, vehicle + VU0463271 + midazolam 5.3 s. *** indicates $p = 0.00146$ (median regression analysis), $n = 6$ animals/group. Transverse bars indicate the median; dots indicate individual values.

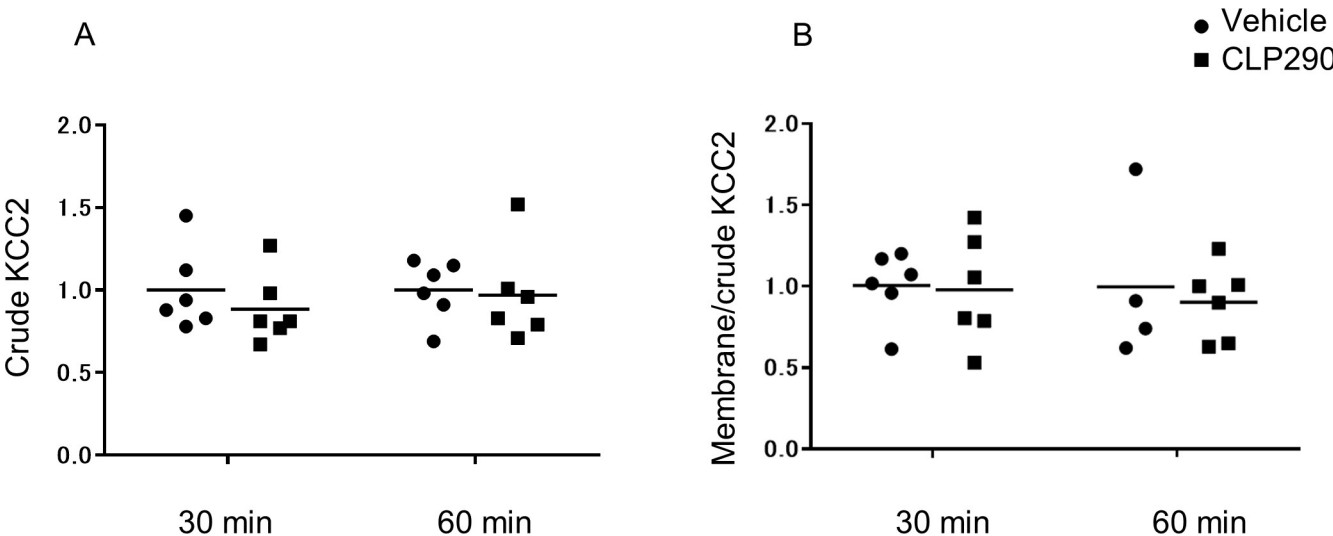

**Fig 6. CLP290 does not alter the protein expression and intracellular distribution of KCC2 in the neonatal cortex.** (A) KCC2 protein expression in the total fraction in the cortex of neonatal rats at 30 min and 60 min after CLP290 administration. 30 min: vehicle 1.0, CLP290 0.89, $p$ = 0.29 (Mann-Whitney U test), $n$ = 6 animals/group. 60 min: vehicle 1.0, CLP290 0.97, $p$ = 0.59 (Mann-Whitney U test), $n$ = 6 animals/group. (B) Ratio of KCC2 protein expression of membrane fraction to total fraction in the cortex of neonatal rats at 30 min and 60 min after CLP290 administration. 30 min: vehicle 1.0, CLP290 0.98, $p$ = 0.93 (Mann-Whitney U test). 60 min: vehicle 1.0, CLP290 0.90, $p$ = 0.91 (Mann-Whitney U test), $n$ = 6 animals/group. Transverse bars indicate the mean; dots indicate individual values.

pretreatment. Treatment with CLP290 30 min and 60 min prior to tissue collections changed neither the crude expression of the KCC2 (Fig 6A) nor its relative membrane expression (Fig 6B). These results suggest that CLP290 promotes the sedative action of midazolam in a way other than affecting the KCC2 expression in the neonatal brain.

## Discussion

We previously assessed the balance between different cation-chloride cotransporter isoforms in immature neurons and found that reducing intracellular chloride ion levels through NKCC1 antagonism facilitates midazolam-induced sedation in neonatal rats [12]. In the current study, we examined whether activation of KCC2, which exerts an opposite effect to that of NKCC1, also promotes midazolam-induced sedation in neonatal rats. A novel KCC2 activator, CLP290, was found to promote the sedative effect of 20 mg/kg midazolam in neonatal rats. In adult rats, CLP290 did not further enhance the sedative effect of midazolam. To elucidate the biochemical basis of these behavioral phenotypes, we performed immunohistochemistry, showing that the combination of CLP290 and midazolam reduced p-CREB expression in the somatosensory cortex in PND7 neonates. This result suggested that changes in neural activity in this region might be involved in the promotion of midazolam-induced sedation by CLP290. Furthermore, VU0463271, a selective KCC2 inhibitor, canceled out the CLP290 effect on midazolam-induced sedation, suggesting that the sedation-promoting action of CLP290 was mediated by the KCC2 pathway. In the previous study, CLP290 increased the intracellular chloride ions via activation of KCC2 in the adult animals showing the reduction of KCC2 expression [13]. The increased intracellular chloride ions by CLP290 has been shown to suppress neuronal overactivations, which effects are expected to exert therapeutic effects in a variety of neurological disease models such as spinal cord injury [15], neuropathic pain [13,16], ischemic seizures [10], nicotine-induced ethanol overdose [17], and streptozotocin-induced diabetes mellitus [22]. However, as far as we know, the effect of CLP290 in naive neonatal rats has not

been described yet, and this is the first report demonstrating that CLP290 enhances the sedative effect of midazolam in neonatal rats.

Midazolam, a positive allosteric modulator of the GABAA receptor, has been shown to have a significantly higher $ED_{50}$ in PND7 neonatal rats than in adult animals, which is consistent with previous studies [9]. Similar to our previous report, we confirmed that 20 mg/kg midazolam had a sedative effect in adult, but not in neonatal rats [12]. In the cerebral cortex of neonatal rats, NKCC1 expression levels are usually highest around PND5-7, and later, KCC2 expression levels gradually increase and become predominant [1]. The dominant expression of NKCC1 in neonates introduces chloride ions into the cytoplasm in contrast to the KCC2 which mediates chloride ion excretion, leading to maintenance of higher $[Cl^-]i$ compared to mature neurons [24,25]. This change in chloride ion homeostasis can account for the different midazolam effects on sedation. Our previous studies also reported that midazolam administration in neonatal rats induces neuronal excitation in the hippocampus, but prior administration of bumetanide, an NKCC1 antagonist, suppresses the excitation and promotes sedation [12,19]. In the current study, administration of CLP290 prior to midazolam could prolong the righting reflex latency, suggesting that CLP290 had a similar effect as bumetanide on midazolam-induced sedation in neonatal rats. Furthermore, CLP290 administration without midazolam did not increase the righting reflex latency. These findings suggest that KCC2 activation by CLP290 promotes the sedative action of midazolam whereas CLP290 itself does not exert sedative effects.

To elucidate the brain regions involved in the promoting effect of CLP290 on midazolam-induced sedation, the expression levels of p-CREB in the cerebral cortex and thalamus of PND7 rats were assessed [12]. Although administration of 20 mg/kg midazolam alone in neonates did not alter the p-CREB expression in the somatosensory cortex and thalamus, midazolam administration to neonatal rats pretreated with CLP290 significantly decreased the number of p-CREB-positive cells in the somatosensory cortex, but not in the thalamus, compared to midazolam alone. These results were consistent with the results of the righting reflex test. The change in p-CREB expression can be used as a surrogate marker reflecting neuronal activity; therefore, the promoting effect of CLP290 on midazolam-induced sedation may be associated with decreased neural activity in the somatosensory cortex of neonates. Furthermore, the brain region where CLP290 acts may be revealed by regional differences in KCC2 expression at PND7. The limited knowledge about the timing of switching from NKCC1 to KCC2 dominance during development shows that the dominance of KCC2 expression over NKCC1 begins from caudal to rostral [1,6,24,25]. The expression of KCC2 in mice starts to increase strongly from embryonic day 14.5 at the thalamus [5,8,26], whereas it is still suppressed at PND7 in the cerebral cortex [27]. Moreover, the expression level of KCC2 in the cerebral cortex in PND7 rats is reported to be less than 10% of that in adult animals [5,7,8,26]. The balance between KCC2 and NKCC1 expression has been suggested to decide the chloride equilibrium potential. It has been shown in neonatal neurons around PND7-10 that GABA activation induces excitation in the cerebral cortex, but inhibition in the thalamus [6]. Moreover, phenobarbital coadministered with bumetanide suppresses epileptiform activity in the cerebral cortex, but not in the thalamus, suggesting that how bumetanide modifies the effect of phenobarbital depends on differences in the maturation of the cation-chloride cotransporter of cortical and subcortical structures [6]. Therefore, the regional differences in p-CREB expression in response to CLP290 administration may be related to differences in KCC2 expression levels and the GABA equilibrium potential at PND7.

While CLP290 has been used as a KCC2 activator in some papers, CLP290 was also reported to enhance GABAA receptor function in a KCC2-independent manner [28]. However, this study examined adult rats and did not consider neonatal rats. To examine this effect

in neonates, we assessed whether the KCC2-selective antagonist VU0463271 suppresses the CLP290 effects in neonatal rats. VU0463271 is a selective KCC2 inhibitor [23] and has been shown to depolarize the GABA equilibrium potential due to inhibition of KCC2 function when administered to adult and neonatal rodents [10,22,29]. In addition, Spoljaric et al. have shown that VU0463271 administration enhances giant depolarizing potentials in neonatal rats and wild-type KCC2$^{+/+}$ mice but does not affect these potentials in KCC2$^{-/-}$ mice [30]. These findings prove sufficiently the specificity of VU0463271 to KCC2. The current study found that combined treatment with CLP290, VU0463271, and midazolam in neonatal rats did not prolong the righting reflex latency. However, VU0463271 alone did not affect midazolam-induced sedation in the absence of CLP290 pretreatment. These results indicate that the enhancing effect of CLP290 on midazolam-induced sedation is prevented by VU0463271 by antagonizing KCC2 activity.

The molecular mechanism underlying the effect of CLP290 on KCC2 in neonatal rats was also examined. Previous studies have reported that CLP257 administration causes transloca-tion of KCC2 to the cell membrane of neurons in the spinal cord in a rat model of neuropathic pain [13]. Another study reported that incubation of hippocampal slices obtained from KCC2-suppressed amyloid precursor protein knockout (*APP$^{-/-}$*) mice with CLP257 increases the membrane KCC2 expression. Furthermore, chronic intraperitoneal CLP290 administra-tion to *APP$^{-/-}$* mice restores KCC2 expression levels in the hippocampus [31]. Based on these findings, it was suggested that CLP290 may translocate KCC2 to the membrane of neurons in the neonatal cerebral cortex. However, their study did not show that CLP290 alters the mem-brane or crude KCC2 expression in the cerebral cortex of PND7 rats. We can propose another possibility that CLP290 alter the phosphorylation levels of KCC2 which are essential for KCC2 function [3,7,32]. Dephosphorylation of Serine$^{940}$ inactivates KCC2-mediated ion transport and increases endocytosis of KCC2 [33]. Phosphorylation of Threonine$^{906}$ and Threonine$^{1007}$ strongly inhibit the transporter activity of KCC2 [34]. Inhibitions of KCC2 phosphorylation at Threonine$^{906}$ and Threonine$^{1007}$ cause depolarizing action of GABA upon binding to GABA receptors during development [35]. However, we have not observe any altered phosphoryla-tion and dephosphorylation in CLP290-treated rats.

Finally, to investigate the physiological effect of CLP290, the effect of CLP290 in neonates was compared with that in adults which showed the predominance of KCC2 over NKCC1. Sul-livan et al. recently reported that CLP290 recovers the decreased KCC2 expression in brain induced by carotid artery ligation in PND7 mice [10]. On the other hand, in mice at PND10, whose KCC2 expression was two-fold compared to those at PND7, the same carotid artery ligation procedure did not change KCC2 expression; furthermore, administration of CLP290 at PND10 did not further increase the total KCC2 expression from base level [12]. These results suggest that the predominance in number, as well as in function, between NKCC1 and KCC2 influences the efficacy of CLP290. In the current study, CLP290 did not promote the sedative action of midazolam at the ED$_{20}$ dose in 4-week-old adult rats, which is consistent with the findings presented by Sullivan et al. [10]. We have not detect any change in the amount or phosphorylation level of KCC2 proteins upon CLP290 treatment, but it remains possible that CLP290 acts via yet unknown phosphorylation sites or other modifications responsible for functional changes in neonates. It may be hypothesized that the resistance to midazolam-induced sedation at 20 mg/kg in neonatal rats is due to their lower KCC2 activity compared to adults. This suggests that the KCC2 in neonates has the capacity for activity enhancement by CLP290, whereas in adults, this capacity is saturated.

This study has several limitations as follows. First, regarding the dose of midazolam, the neonatal rats were sedated after midazolam administration at doses greater than 20 mg/kg without CLP290 treatment. A possible hypothesis is that higher midazolam doses might

severely reduce neuronal activity in the thalamus and its neural output to the cerebral cortex, leading to a sedative state. Second, we could not examine changes in the intracellular chloride ion levels after CLP290 administration due to technical difficulties. However, the result that KCC2 activation using CLP290 enhanced the effect of midazolam which can be generally accounted for by the activation of the GABAA receptor suggested indirectly decreased intracellular chloride ion levels upon CLP290 administration. Third, the mechanism underlying the CLP290-mediated enhancement of KCC2 function could not be identified. Some KCC2 phosphorylation sites have been identified but the function of most sites remains unknown in neonates. Moreover, we cannot rule out the possibility of other phosphorylation sites or different protein modifications.

Clinically, midazolam is usually administered to infants and children for sedation during operation and intensive care, and anesthetists face the problem that some children are resistant to sedation with midazolam [36]. The combination of midazolam with CLP290 might be a promising option for overcoming this serious problem. Our findings suggest that KCC2 activation by CLP290 promotes safe sedation with midazolam and may provide a breakthrough for midazolam-resistant sedation in clinical settings.

## Supporting information

**S1 Checklist. ARRIVE guidelines checklist.**
(PDF)

**S1 Fig. Coadministration of midazolam with CLP290 does not change p-CREB expression in the neonatal thalamus.** (A-D)Representative pictures of p-CREB expression in the neonatal ventral thalamus after the following administration protocols: (A) vehicle + saline, (B) vehicle + midazolam, (C) vehicle + midazolam, (D) CLP290 + midazolam. (E) Mean numbers of p-CREB-positive cells for the four experimental groups: vehicle + saline 84.46 ($n$ = 9), vehicle + midazolam 91.18 ($n$ = 8), CLP290 + midazolam ($n$ = 6), CLP290 + midazolam 55.19 ($n$ = 7). Transverse and vertical bars indicate the mean and SE, respectively; dots indicate individual values.
(TIF)

**S1 File.**
(XLSX)

## Acknowledgments

The authors would like to thank Ms. Akiko Adachi and Ms. Yuki Yuba (Research Assistants, Department of Anesthesiology and Critical Care Medicine, Yokohama City University Graduate School of Medicine, Yokohama, Japan) for excellent technical support and animal care.

## Author Contributions

**Conceptualization:** Tomoyuki Miyazaki.

**Formal analysis:** Takahiro Mihara.

**Investigation:** Akiko Doi, Maiko Ikeda, Ryo Niikura.

**Methodology:** Akiko Doi, Tomoyuki Miyazaki.

**Project administration:** Akiko Doi.

**Resources:** Akiko Doi, Tomoyuki Miyazaki, Takahiro Mihara, Maiko Ikeda, Ryo Niikura, Tomio Andoh, Takahisa Goto.

**Supervision:** Tomio Andoh, Takahisa Goto.

**Writing – original draft:** Akiko Doi, Takahiro Mihara.

**Writing – review & editing:** Tomoyuki Miyazaki, Tomio Andoh.

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
