## [Decision Letter · Decision Letter 0]

15 May 2020

PONE-D-20-09548

CLP290 promotes the sedative effects of midazolam in neonatal rats in a KCC2-dependent manner: A laboratory rat study

PLOS ONE

Dear Dr. Miyazaki,

Thank you for submitting your manuscript to PLOS ONE. After careful consideration, we feel that it has merit but does not fully meet PLOS ONE’s publication criteria as it currently stands. Therefore, we invite you to submit a revised version of the manuscript that addresses the points raised during the review process. Note also that statistical analysis was not performed in an appropriate manner. Particularly, repeated measures analysis of variance and two-way analysis of variance were not considered, respectively, in figures 1 and S1. This is only an example, but all your statistical approach requires a reappraisal. Details on fixation are also missing from the manuscript. It is well known that pups require a different procedure when compared to adults to obtain an optimal brain preservation.

We would appreciate receiving your revised manuscript by Jun 27 2020 11:59PM. To enhance the reproducibility of your results, we recommend that if applicable you deposit your laboratory protocols in protocols.io, where a protocol can be assigned its own identifier (DOI) such that it can be cited independently in the future. For instructions see: http://journals.plos.org/plosone/s/submission-guidelines#loc-laboratory-protocols

We look forward to receiving your revised manuscript.

Kind regards,

Giuseppe Biagini, MD

Academic Editor

PLOS ONE

Journal Requirements:

2. Thank you for including your ethics statement:  "all animal care procedures in accordance with the standards approved by the Yokohama City University Institutional Animal Care and Use Committee (approval number; #F-A-16-045)."

Please amend your current ethics statement to confirm that your named ethics committee specifically approved this study.

For additional information about PLOS ONE submissions requirements for ethics oversight of animal work, please refer to http://journals.plos.org/plosone/s/submission-guidelines#loc-animal-research  

3. In your methods section, please provide the catalog numbers of CLP290 and VU0463271.

4. To comply with PLOS ONE submissions requirements, in your Methods section, please provide additional information on the animal research and ensure you have included details on (1) the total number of rats used in this study, (2) how often animals were checked on for welfare, and (3) whether decapitation was performed by trained personnel.

5. In your methods section, please provide the full and detailed criteria used to determine righting reflex latency times to ensure that other researchers can replicate and reproduce your experiments.

Reviewers' comments:

Reviewer's Responses to Questions

**Comments to the Author**

1. Is the manuscript technically sound, and do the data support the conclusions?

Reviewer #1: Partly

Reviewer #2: Partly

2. Has the statistical analysis been performed appropriately and rigorously? 

Reviewer #1: Yes

Reviewer #2: I Don't Know

3. Have the authors made all data underlying the findings in their manuscript fully available?

Reviewer #1: Yes

Reviewer #2: Yes

4. Is the manuscript presented in an intelligible fashion and written in standard English?

Reviewer #1: No

Reviewer #2: Yes

5. Review Comments to the Author

Reviewer #1: The study by Akiki Doi et al “CLP290 promotes the sedative effects of midazolam in neonatal rats in a KCC2-dependent manner: A laboratory rat study” provides evidence that activation of the KCC2 cotransporter isoform can facilitate midazolam—triggered sedation. This reviewer has a number of comments/questions to authors.

Abstract. In the context of their results, what authors mean by saying: Contrastingly, CLP290 alone did not enhance the sedative effect of midazolam?

Introduction. “GABAA receptor activation exhibits excitatory toxicity…” Membrane depolarization, usually leading to neuronal excitation, is a normal physiological event, not necessarily neurotoxic.

Introduction. “there has been no study showing that these compounds inhibit GABAA receptor-induced excitation in immature neurons…” Present study also does not show this. What is shown is facilitation of benzodiazepines sedative effects. CLP290 alone does not do anything. It might change conformation of benzodiazepine binding site at GABAA receptor, directly or indirectly, in such a way that midazolam could change ion selectivity of the associated ion channel, or block it, for example, or whatever else. However, all these possibilities are too speculative. Omitting ungrounded speculations might benefit the present manuscript.

Introduction. “p-CREB immunohistochemistry to determine the neural circuits underlying the sedative effect of CLP290…” pCREB is a transcriptional factor, which can be used as a nonspecific marker for protein synthesis, perhaps, neuronal activation, but not of neuronal excitation or excitability (here and throughout the manuscript). Of course, this immunohistochemistry may not “determine neuronal circuits". Please, correct these scientific/linguistic inconsistencies.

Drugs. “…dissolved CLP290 in 10% 2-hydroxypropyl-β-cyclodextrin or suspended it in 0.5% carboxymethyl cellulose with a final concentration of 0.66% CLP290.” Why two different solvents (both valid) were used and in which experiments?

Immunohistochemistry protocol. “At 75 min after midazolam or saline injection, the rats were deeply anesthetized…” Why this time interval has been chosen?

Immunohistochemistry protocol. Indicate how many times the sections were washed between all steps (not "several times).

Immunohistochemistry protocol. “…(100 × 200 μm2) slices and thalamus (200 × 500 μm2)… This is not very clear and even confusing. Please, extend this description sufficiently for readers to understand what it means and to be able to replicate this sampling technique.

Biochemistry. “Next, 1 mm brain slices were obtained using a brain slicer and the somatosensory cortex was extracted. Subsequently, the slices were incubated…” Please, explain how the somatosensory area was found and “extracted”. Not clear – only these extracted parts were than used or remaining slices.

Statistics. Why authors opted for Mann-Whitney test instead of parametric ANOVA?

Immunohistochemistry results and figure 2 legend. Statistics was done for “N = 9-12 slices/group”. This is not valid. The numbers obtained from all sections of a single animal should be averaged. N (number of rats) in each group should be given. I understand that sample sizes for some groups might be too small for statistical comparisons. Should be this the case, authors should consider the possibility of either omitting this part or removing the entire submission until the data are complete.

Why “brain CLP290 levels were expected to have a slower increase than those of VU0463271”?

Discussion. Authors compare the results obtained in the somatosensory cortex, thalamus and hippocampus, but I was not able to find data for the thalamus and hippocampus in the manuscript.

Discussion. The hypothesis of distinct switching timing for cation-chloride cotransporters appears somewhat immature. It should be better explained and referenced. If not possible – omitted.

There is an excessive use of phrases starting with “We…”. This makes the story a bit subjective. Although personal statements of interest and of hypotheses cannot be fully circumvented in scientific papers, the use of sentences written in the third person is preferential.

Reviewer #2: In the current Ms, the authors want to investigate whether CLP290 promotes the sedative effects of midazolam in neonatal rats via an action on KCC2.

This Ms is written in a clear structure and has some substantial interest. However, I have some concerns listed below.

1) In P6, the authors mentioned that they used bentobarbital. Did the authors notice if the pentobarbital anesthetic effect was altered by the pharmacological drugs used in their study? Which dose of sodium pentobarbital has been used?

2) To prove further their concept, the authors should perform some experiments showing the effect of midazolam on adult rat treated with VU0463271 or CLP290 during their behavioral experiments.

3) In order to be able to state in the title of their Ms that CLP290 has an effect via an action on KCC2 I am expecting to see more convincing arguments or at least more direct evidences that KCC2 is involved.

4) It would be good to provide supplemental figures in a more standard format.

I have some few comments regarding the methods section:

1) It would be interesting for the reader not familiar to the field to give a description (even brief) of the righting reflex test.

2) Ln107: please could the authors explain why they disssolved CLP290 in 2 different vehicles? Which vehicle was used for which experiments?

3) Ln161: Could the authors explain why they used isoflurane this time and not pentobarbital? I presume that the rats were sacrified before removing the brain, please could you give more details regarding this?

4) Ln162-164: Please could you describe your dissection buffer and your ACSF?

5) Ln163: Please give reference of the vibroslicer.

The authors should check carefully the numerous typos or confusions present in the text, I have listed few of them below:

1) Abstract ln29: Therefore, midazolam, a benzodiazepine agonist: I presume that you mean benzodiazepine receptor agonist.

2) ln30 "This indicates that the balance between the NKCC1 and KCC2 is important for GABA agonist action". I don't think this conclusion was drawn from the midazolam experiments.

3) Abstract, Ln37: « Contrastingly, CLP290 alone did not enhance the sedative effect of midazolam.” Not clear at all. What do you mean exactly? I guess you meant that CLP290 alone had no sedative effect?

4) ln59: "higher intracellular [Cl-]I levels": I presume it should not be a capital I.

5) Ln198 Strangely the legend of the figure 1 is at the end of the Methods section.

6) ln255: please be consistent and use capital letters for CLP290

7) ln311: “To our knowledge, this is the first study to report that CLP290 modulates GABAA receptor-mediated pharmacology”. I think this statement is not correct. CLP290 does not interfere with pharmacology at GABAA receptor but affect indirectly the effect of GABAA receptor activation.

8) ln379 "“A possible resulting hypothesis is that higher midazolam doses might deeply attenuate neuronal activity in the thalamus deeply,”. Deeply is repeated twice.

6. PLOS authors have the option to publish the peer review history of their article (what does this mean?). If published, this will include your full peer review and any attached files.

Reviewer #1: No

Reviewer #2: No

---

## [Author Response · Author response to Decision Letter 0]

15 Jan 2021

Response to Reviewers

Response to PLoS One editor:

1. Note also that statistical analysis was not performed in an appropriate manner. Particularly, repeated measures analysis of variance and two-way analysis of variance were not considered, respectively, in figures 1 and S1. This is only an example, but all your statistical approach requires a reappraisal. 

Thank you for your suggestion. As you mentioned, the ANOVA test is usually considered for parametric data. However, our behavioral data in this study did not have a normal distribution according to the test of normality and, therefore, the ANOVA should not be used. In this case, quantile regression is suitable because it can be used without an assumption of a specific distribution, and we performed the statistical analysis using quantile regression in the following results (Fig 2B, Fig 3B, and Fig 5B).

2. Details on fixation are also missing from the manuscript. It is well known that pups require a different procedure when compared to adults to obtain an optimal brain preservation.

Thank you for your comment. As you mentioned, the antigenicity in young brains decreases with overfixation. We have already optimized the fixation protocol for neonates, as shown in our previous paper [12], and this study used the same method. We added details of the fixation method to the methods section (Lines 152-156).

3. To enhance the reproducibility of your results, we recommend that if applicable you deposit your laboratory protocols in protocols.io, where a protocol can be assigned its own identifier (DOI) such that it can be cited independently in the future. For instructions see: http://journals.plos.org/plosone/s/submission-guidelines#loc-laboratory-protocols

Thank you for your suggestion. We will try this procedure after this paper gets accepted.

4. To comply with PLOS ONE submissions requirements, in your Methods section, please provide additional information on the animal research and ensure you have included details on (1) the total number of rats used in this study, (2) how often animals were checked on for welfare, and (3) whether decapitation was performed by trained personnel.

Thank you for your comment. Following your instruction, we included the following information in our manuscript: (1) The total numbers of rats used were added to the methods section (Line 97); (2) all rats were checked for their well-being daily at the time of feeding (Line 101); and (3) we declared in the methods section that trained personnel performed the decapitation (Line 177).

5. In your methods section, please provide the full and detailed criteria used to determine righting reflex latency times to ensure that other researchers can replicate and reproduce your experiments.

Thank you for your comment. We added details of this method and criteria for determining the righting reflex latency to the methods section (Lines 115-119).

Reviewer #1: The study by Akiki Doi et al “CLP290 promotes the sedative effects of midazolam in neonatal rats in a KCC2-dependent manner: A laboratory rat study” provides evidence that activation of the KCC2 cotransporter isoform can facilitate midazolam—triggered sedation. This reviewer has a number of comments/questions to authors.

Abstract. In the context of their results, what authors mean by saying: Contrastingly, CLP290 alone did not enhance the sedative effect of midazolam?

Thank you for your comment. We wanted to describe that “By contrast, CLP290 alone did not exert sedative effects ” and rephrased the corresponding sentence (Line 27).

Introduction. “GABAA receptor activation exhibits excitatory toxicity…” Membrane depolarization, usually leading to neuronal excitation, is a normal physiological event, not necessarily neurotoxic.

Thank you for your correction, and we fully agree with it. We rephrased the corresponding sentence (Line 61).

Introduction. “there has been no study showing that these compounds inhibit GABAA receptor-induced excitation in immature neurons…” Present study also does not show this. What is shown is facilitation of benzodiazepines sedative effects. CLP290 alone does not do anything. It might change conformation of benzodiazepine binding site at GABAA receptor, directly or indirectly, in such a way that midazolam could change ion selectivity of the associated ion channel, or block it, for example, or whatever else. However, all these possibilities are too speculative. Omitting ungrounded speculations might benefit the present manuscript.

Thank you for your comment with which we agree. In this study, we did not perform experiments directly showing that GABAA receptor activation induces neuronal excitation in neonates. Throughout our manuscript, we suggest that activation of KCC2 promotes the sedative effect of GABAA receptor activation in neonates. We rephrased the corresponding sentence (Lines 80-81).

Introduction. “p-CREB immunohistochemistry to determine the neural circuits underlying the sedative effect of CLP290…” pCREB is a transcriptional factor, which can be used as a nonspecific marker for protein synthesis, perhaps, neuronal activation, but not of neuronal excitation or excitability (here and throughout the manuscript). Of course, this immunohistochemistry may not “determine neuronal circuits". Please, correct these scientific/linguistic inconsistencies.

Thank you for your comment. We agree with the concerns expressed here. As you mentioned, we used p-CREB as a non-specific marker for neuronal activation, not excitation. That is to say, we performed p-CREB immunostaining to identify brain regions related to the promoting effect of CLP290 on midazolam-induced sedation, not to determine neuronal circuits. We rephrased the corresponding sentences (Lines 84-86).

Drugs. “…dissolved CLP290 in 10% 2-hydroxypropyl-β-cyclodextrin or suspended it in 0.5% carboxymethyl cellulose with a final concentration of 0.66% CLP290.” Why two different solvents (both valid) were used and in which experiments?

Thank you for this constructive question. In neonatal behavioral tests and immunostaining experiments, we used CLP290 kindly supplied by Dr. De Koninck. He indicated this drug had to be dissolved in hydroxypropyl-β-cyclodextrin (HPβCD). Two years ago, we were not able to contact him and, therefore, bought CLP290 from another company. CLP290 has very low solubility in water, and we could not find a way to dissolve it even in HPβCD. We consulted a chemist to find a way to dissolve it and finally purchased CLP290 from Sigma. Their drug can be dissolved in 5% carboxy methylcellulose, and we confirmed the same efficacy in behavioral experiments of this drug as that supplied by Dr. De Koninck. The experiments using VU0463271, adult behavioral tests, and western blot experiments were performed using CLP290 purchased from Sigma.

Immunohistochemistry protocol. Immunohistochemistry protocol. “At 75 min after midazolam or saline injection, the rats were deeply anesthetized…” Why this time interval has been chosen?

Thank you for your comment. Accordingly, we corrected this sentence. We have performed immunohistochemistry using the sample obtained from rats at 45 min after administration of midazolam or saline, as previously published [12]. (Line 149)

Immunohistochemistry protocol. Indicate how many times the sections were washed between all steps (not "several times).

Thank you for your suggestion. We added the exact number of washing steps to the appropriate sentences in the methods section (Lines 161, 164, 187).

Immunohistochemistry protocol. “…(100 × 200 μm2) slices and thalamus (200 × 500 μm2)… This is not very clear and even confusing. Please, extend this description sufficiently for readers to understand what it means and to be able to replicate this sampling technique.

Thank you for this valuable suggestion. We added the detailed methodology of sampling for the immunohistochemistry staining to the methods section (Lines 156-157, 167-172).

Biochemistry. “Next, 1 mm brain slices were obtained using a brain slicer and the somatosensory cortex was extracted. Subsequently, the slices were incubated…” Please, explain how the somatosensory area was found and “extracted”. Not clear – only these extracted parts were than used or remaining slices.

Thank you for your comment. We added the details regarding the method to identify the somatosensory region to the methods section (Lines 180-183).

Statistics. Why authors opted for Mann-Whitney test instead of parametric ANOVA?

Thank you for your comment. The righting reflex test is usually analyzed using a nonparametric test due to its non-normal distribution. Furthermore, our results obtained from biochemical experiments also showed non-normal distribution. Therefore, we performed median regression analysis in righting reflex experiment because median regression analysis does not require an assumption of a specific distribution and suitable for them, and used Mann-Whitney analysis in biochemical experiment (Lines 213-218, 221-224).

Immunohistochemistry results and figure 2 legend. Statistics was done for “N = 9-12 slices/group”. This is not valid. The numbers obtained from all sections of a single animal should be averaged. N (number of rats) in each group should be given. I understand that sample sizes for some groups might be too small for statistical comparisons. Should be this the case, authors should consider the possibility of either omitting this part or removing the entire submission until the data are complete.

Thank you for your comment with which we fully agree. In the initial manuscript, we analyzed the immunohistochemical data by indicating with n the number of slices, which is not a valid approach. In the revised manuscript, we performed additional experiments to add animals to each group and analyzed the data such that n indicates now the number of animals, and analyzed using the Tukey-Kramer test (Fig 4, S1 Fig and Lines 219-220).

Why “brain CLP290 levels were expected to have a slower increase than those of VU0463271”?

Thank you for your comment. We mentioned that VU0463271 was administered intraperitoneally, whereas CLP290 was administered orally. Therefore, it is expected that compared to VU0463271, the concentration of CLP290 in the brain, as well as in the blood, increases slower. We added a sentence to explain this (Lines 309-310).

Discussion. Authors compare the results obtained in the somatosensory cortex, thalamus and hippocampus, but I was not able to find data for the thalamus and hippocampus in the manuscript. 

Thank you for your comment. In this study, we focused on the thalamocortical pathway known to be related to sedation. Therefore, the description regarding the hippocampus was unnecessary and has been deleted. We added representative images and a graph showing the number of p-CREB-positive neurons in the thalamus to S1 Fig and also revised the corresponding figure legend (Lines 612-618).

Discussion. The hypothesis of distinct switching timing for cation-chloride cotransporters appears somewhat immature. It should be better explained and referenced. If not possible – omitted.

Thank you for your comment. In our manuscript, we added some references suggesting the following lines of evidence (Lines 400-414): 1. the expression of KCC2 in mice is still lowere at postnatal day (PND) 7 in the cerebral cortex [27] ,whereas in that in the thalamus increases from embryonic day 14 [5, 8, 26] and become almost comparable with adults at PND0 [7]; 2. In neonatal neurons around PND7-10, GABA receptor activation induces neuronal activation in the cerebral cortex, whereas GABA receptor activation induces neuronal inhibition in the thalamus, similar to that in adults [6]. These reports are consistent with our p-CREB experiments showing that administration of midazolam with activation of KCC2 by CLP290 reduced neuronal activation in the cortex but no change was observed in the thalamus even though with CLP290 administration . Our hypothesis is that the regional differences in p-CREB expression in response to CLP290 administration may be related to differences in KCC2 expression levels and the GABA equilibrium potential at PND7. To strengthen this hypothesis, further research is needed in the future.

There is an excessive use of phrases starting with “We…”. This makes the story a bit subjective. Although personal statements of interest and of hypotheses cannot be fully circumvented in scientific papers, the use of sentences written in the third person is preferential.

Thank you for your suggestion. We rephrased these sentences accordingly.

Reviewer #2: In the current Ms, the authors want to investigate whether CLP290 promotes the sedative effects of midazolam in neonatal rats via an action on KCC2.

This Ms is written in a clear structure and has some substantial interest. However, I have some concerns listed below.

1) In P6, the authors mentioned that they used pentobarbital. Did the authors notice if the pentobarbital anesthetic effect was altered by the pharmacological drugs used in their study? Which dose of sodium pentobarbital has been used?

Neonatal rats were intraperitoneally administered 342 mg/kg of pentobarbital (Somnopentil, 64.8 mg/ml, Kyoritsu Seiyaku Corp, Tokyo, Japan) (Line 151), and with this dose, all rats fell asleep within 2 min. Since an excess dose of 10 times the dose required for sedation in adult was administered, the synergistic effect with CLP290 and midazolam could not be directly evaluated. 

2) To prove further their concept, the authors should perform some experiments showing the effect of midazolam on adult rat treated with VU0463271 or CLP290 during their behavioral experiments.

Additional experiments were conducted to investigate whether CLP290 enhances the sedative effect of midazolam in adult rats, as well as in neonatal rats (Lines 120-133). As shown in Fig 1, a dose-response curve for midazolam was created for both neonatal and adult rats, showing that the 20% effective dose (ED20), equivalent to 20 mg/kg of midazolam in neonates, was 8.6 mg/kg in adult rats (Lines 239-241). Thus, 2 h after oral CLP290 administration, midazolam 8.6 mg/kg was intraperitoneally administered to adult animals, followed by the righting reflex test. In contrast to neonatal rats, CLP290 at 100 mg/kg exerted little influence on the sedative midazolam effect at ED20 (Fig 3 and Lines 247-251). The CLP290 effect was not statistically significant and, therefore, we did not perform subsequent experiments using VU0463271 in adult animals. 

3) In order to be able to state in the title of their Ms that CLP290 has an effect via an action on KCC2 I am expecting to see more convincing arguments or at least more direct evidence that KCC2 is involved.

Thank you for your comment. To date, CLP290 has been used as a KCC2 activator in the following disease models: spinal cord injury [15], neuropathic pain [13, 16], ischemic seizures [10], nicotine-induced ethanol overdose [17], and streptozotocin-induced diabetes [22]. Furthermore, VU0463271, a specific KCC2 inhibitor, is available (Spoljaric et al. showed that VU0463271 enhances giant depolarizing potentials in neonatal rats and wild-type KCC2+/+ mice but does not affect these potentials in KCC2-/- mice, suggesting that the VU0463271 effect is KCC2-dependent [30]). Our study also showed that antagonization of KCC2 using VU0463271 attenuated the promoting effect of CLP290 on midazolam-induced sedation, suggesting that the effect of CLP290 is directly or indirectly mediated by KCC2 (Lines 417-427).

4) It would be good to provide supplemental figures in a more standard format.

Thank you for your comment. According to your suggestion, and we have revised the supplemental figure. 

I have some few comments regarding the methods section:

1) It would be interesting for the reader not familiar to the field to give a description (even brief) of the righting reflex test.

Thank you for your suggestion. We added details of this method and criteria for determining the righting reflex latency to the methods section (Lines 115-119).

2) Ln107: please could the authors explain why they disssolved CLP290 in 2 different vehicles? Which vehicle was used for which experiments?

Thank you for this constructive question. In neonatal behavioral tests and immunostaining experiments, we used CLP290 kindly supplied by Dr. De Koninck. He indicated this drug had to be dissolved in hydroxypropyl-β-cyclodextrin (HPβCD). Two years ago, we were not able to contact him and, therefore, bought CLP290 from another company. CLP290 has very low solubility in water, and we could not find a way to dissolve it even in HPβCD. We consulted a chemist to find a way to dissolve it and finally purchased CLP290 from Sigma. Their drug can be dissolved in 5% carboxy methylcellulose, and we confirmed the same efficacy in behavioral experiments of this drug as that supplied by Dr. De Koninck. The experiments using VU0463271, adult behavioral tests, and western blot experiments were performed using CLP290 purchased from Sigma.

3) Ln161: Could the authors explain why they used isoflurane this time and not pentobarbital? I presume that the rats were sacrificed before removing the brain, please could you give more details regarding this?

Thank you for your comment. As you mentioned, the rats were sacrificed before removing the brains. In this biochemical experiment, some animals were processed simultaneously for brain sample collection, and we considered that anesthesia using a desiccator filled with isoflurane was easier for achieving this purpose. (Lines 176-177)

4) Ln162-164: Please could you describe your dissection buffer and your ACSF?

Thank you for your comment. The compositions of the dissection buffer (Lines 178-180) and ACSF (Lines 184-185) were added to the manuscript.

5) Ln163: Please give reference of the vibroslicer.

Thank you for your comment. We added this information to the revised manuscript (Line 181).

The authors should check carefully the numerous typos or confusions present in the text, I have listed few of them below:

1) Abstract ln29: Therefore, midazolam, a benzodiazepine agonist: I presume that you mean benzodiazepine receptor agonist.

Thank you for your comment. We corrected this sentence (Line 21).

2) ln30 "This indicates that the balance between the NKCC1 and KCC2 is important for GABA agonist action". I don't think this conclusion was drawn from the midazolam experiments.

Thank you for your comment, and we agree with you. We revised the part of the abstract related to this sentence (Lines 23-25).

3) Abstract, Ln37: « Contrastingly, CLP290 alone did not enhance the sedative effect of midazolam.” Not clear at all. What do you mean exactly? I guess you meant that CLP290 alone had no sedative effect?

Thank you for your comment. We rephrased this sentence as follows: “By contrast, CLP290 alone did not exert sedative effects” (Line 27).

4) ln59: "higher intracellular [Cl-]I levels": I presume it should not be a capital I.

Thank you for your correction. We corrected this typo (Line 52).

5) Ln198 Strangely the legend of the figure 1 is at the end of the Methods section.

Thank you for your comment. We changed the position of this figure legend in the revised manuscript. 

6) ln255: please be consistent and use capital letters for CLP290

Thank you for your correction. We ensured that CLP290 is spelled consistently throughout the entire manuscript. 

7) ln311: “To our knowledge, this is the first study to report that CLP290 modulates GABAA receptor-mediated pharmacology”. I think this statement is not correct. CLP290 does not interfere with pharmacology at GABAA receptor but affect indirectly the effect of GABAA receptor activation.

Thank you for very much your comment. The proposed mechanism underlying the observed CLP290 effect is to change the [Cl-]i by activating KCC2, followed by indirect modification of the action induced by GABA receptor activation. We changed the manuscript as follows: “ However, as far as we know, the effect of CLP290 in naive neonatal rats has not been described yet, and this is the first report demonstrating that CLP290 enhances the sedative effect of midazolam in neonatal rats.” (Lines 371-373).

8) ln379 “A possible resulting hypothesis is that higher midazolam doses might deeply attenuate neuronal activity in the thalamus deeply,”. Deeply is repeated twice.

Thank you for your correction. Accordingly, we corrected this sentence (Lines 461-463).

Besides response to the comments from reviewers, we changed following sentences to improve readability;

1. Lines 18-23

　　before “therefore, their intracellular chloride ion ([Cl]-i) concentration is higher than that in mature neurons; further, hyperpolarization does not occur due to γ-aminobutyric acid (GABA) receptor activation. Therefore, midazolam, a benzodiazepine agonist, does not exert a sufficient sedative effect in neonates.”

　　after “The intracellular chloride ion concentration ([Cl-]i) is higher in immature neurons than in mature neurons; therefore, γ-aminobutyric acid type A (GABAA) receptor activation in immature neurons does not cause chloride ion influx and subsequent hyperpolarization. In our previous work, we found that midazolam, benzodiazepine receptor agonist, causes less sedation in neonatal rats compared to adult rats and that NKCC1 blockade by bumetanide enhances the midazolam-induced sedation in neonatal, but not in adult, rats.”

2. Lines 33-37

　　before “To our knowledge, this is the first study on the effects of CLP290 in neonates. It provides behavioral and histological evidence that CLP290 improves the sedative effect of GABAergic drugs through KCC2 activation. We expect that the clinical application of CLP290 could provide a breakthrough in midazolam-induced resistance and adverse effects.”

after “To our knowledge, this study is the first report showing the sedation-promoting effect of CLP290 in neonates and providing behavioral and histological evidence that CLP290 reverted the sedative effect of GABAergic drugs through the activation of KCC2. Our data suggest that the clinical application of CLP290 may provide a breakthrough in terms of midazolam-resistant sedation.”

3. Lines 54-56

before “The switch in the functional role of the GABAA receptor from excitatory to inhibitory occurs between PND 8 to 14, which differs depending on the brain region [4-6].”

after “The switch in the functional role of the GABAA receptor from excitatory to inhibitory GABA signaling occurs between PND8 and 14, which correlates with the switch in dominant expression from NKCC1 to KCC2 [6-8].”

4. Lines 82-90

　　before “We hypothesized that KCC2 activation promotes GABAA receptor-induced neuronal inhibition and sedative effects of midazolam in neonatal rats. To test this hypothesis, we conducted loss of righting reflex behavioral tests to examine the supportive effect of CLP290 on midazolam-induced sedation in neonatal animals. Moreover, we conducted p-CREB immunohistochemistry to determine the neural circuits underlying the sedative effect of CLP290. Further, we measured the cell surface expression of KCC2 to investigate the biochemical processes underlying the CLP290 effects in vitro.”

　　after “We hypothesized that activation of KCC2 promotes the sedative effect of GABAA receptor activation in immature rodents. To test this hypothesis, we assessed the righting reflex behavior of neonatal and adult animals treated with midazolam and CLP290. Next, we conducted phosphorylated cyclic adenosine monophosphate-response element-binding protein (p-CREB) immunohistochemistry to determine the brain region related to the CLP290 effect on midazolam-induced sedation. Moreover, to elucidate the underlying mechanism, we investigated whether the sedation-promoting effect of CLP290 in neonates can be antagonized by the KCC2-selective antagonist VU0463271. Finally, we measured the cell surface expression of KCC2 to elucidate the biochemical processes underlying the in vivo effect of CLP290.”

5. Lines 134-143

　　before “3. Neonatal rats were orally administrated with CLP290 at 100 mg/kg or an equal vehicle volume. Subsequently, we intraperitoneally injected VU0463271 or an equal vehicle volume after 20 and 50 min at 20 ml/kg and 10 ml/kg, respectively. Additionally, we intraperitoneally injected midazolam at 20 mg/kg 2 h after CLP290 or vehicle administration.”

　　after “3. To assess the involvement of KCC2 in the promotion effect of CLP290 on midazolam-induced sedation in neonatal rats, rats were pretreated with CLP290 followed by VU0463271 prior to the midazolam administration. That is, neonatal rats were randomly assigned to two groups and orally pretreated with CLP290 (100 mg/kg) or an equal volume of vehicle 2 h prior to the midazolam administration. Subsequently, each group was subdivided into two groups, and VU0463271 or an equal volume of vehicle was injected twice intraperitoneally at 0.02 ml/g and 0.01 ml/g 20 min and 50 min, respectively, after the first pretreatment. All rats received midazolam at 20 mg/kg intraperitoneally 70 min after the second administration of VU0463271 or vehicle. The righting reflex latency of all rats was measured 30 min after administration of midazolam (n = 6).”

6. Lines 145-149

　　before “We conducted the immunohistochemistry experiments with the purpose of assessing the neuronal excitability using the combination of midazolam and CLP290 (experimental outcome). We orally administered neonate and adult rats with CLP290 at 100 mg/kg or an equal vehicle volume 2 h prior to intraperitoneal midazolam injection at 20 mg/kg or an equal saline volume.”

　　after “This immunohistochemistry experiment was conducted to assess the neuronal activity following the separate or combined administration of midazolam and CLP290. Neonatal rats were randomly assigned to two groups and orally treated with CLP290 (100 mg/kg) or an equal volume of vehicle. Subsequently, 2 h after CLP290 or vehicle administration, each group was subdivided into two groups to receive midazolam at 20 mg/kg or an equal volume of saline intraperitoneally.”

7. Lines 241-246

　　before “To examine whether KCC2 activation promoted the sedative effect of midazolam, we administered CLP290 to PND 7 rats. We found that CLP290 administration significantly prolonged the righting reflex latency compared with vehicle administration at 15 min (Fig 1B) and 30 min (Fig 1C) after midazolam injection. Moreover, the administration of CLP290 alone did not have an effect on the righting reflex latency (Fig 1B and 1C). This indicates that KCC2 activation through oral CLP290 administration promoted the sedative effect of midazolam in the neonates while a similar midazolam dose without CLP290 pretreatment had no sedative effect in the neonates unlike the adults.

　　after “In PND7 rats, showing resistance to midazolam-induced sedation, CLP290 pre-administration significantly prolonged the righting reflex latency compared to vehicle at 30 min after midazolam injection (Fig 2B). Moreover, the administration of CLP290 alone had no effect on the righting reflex latency (Fig 2B). This suggests that prior activation of KCC2 by CLP290 might promote the sedative effect of midazolam in neonates whereas CLP290 has no sedative effect by itself.”

8. Lines 279-280

　　before “To examine whether the CLP290-induced sedative effect resulted from an inhibition of neuronal excitation, we conducted p-CREB immunohistochemistry to determine the number of excitatory neurons in awake-related brain regions.”

　　after “To identify brain regions involved in the CLP290 effect on midazolam-induced sedation, p-CREB immunohistochemistry was conducted (Fig 4A).” 

9. Lines 282-292

before “Contrastingly, midazolam administration to PND 7 rats pretreated with CLP290 

significantly decreased the number of p-CREB positive cells compared with vehicle administration, which was consistent with the righting reflex test results (Fig 2B, 2D and 2E). However, there was no significant among-group difference in the number of p-CREB positive cells in the thalamus (S2 Fig). These findings indicated that the administration of 20 mg/kg midazolam alone had an insignificant effect on neuronal status; however, CLP290 pretreatment promoted the inhibitory effect of midazolam in the somatosensory cortex in neonatal rats.”

　　after “By contrast, midazolam administration to neonatal rats pretreated with CLP290 significantly decreased the number of p-CREB-positive cells compared with vehicle administration (Fig 4C, E, and F). Moreover, the administration of CLP290 alone had no effect on the number of p-CREB-positive cells (Fig 4B, D, and F). These findings are inversely correlated to the level of sedation as assessed by the loss of the righting reflex (Fig 2). However, the numbers of p-CREB-positive cells in the thalamus were not significantly different among groups (Supplement Fig1). These findings showed that a single dose of 20 mg/kg midazolam had little effect on neuronal activity in the cerebral cortex and thalamus of PND7 rats. Midazolam administration after pretreatment with CLP290 exerted inhibitory effects on neuronal activity exclusively in the somatosensory cortex of neonatal rats, which is consistent with the behavioral results in the righting reflex test shown in Fig 2.”

10. Lines 317-323

　　before “As aforementioned, the righting reflex latency after midazolam injection with CLP290 pretreatment was significantly longer than that without CLP290; further, VU0463271 co-administration significantly attenuated the effect of CLP290 on the righting reflex (Fig 3B). These results indicated that the CLP290 effect of promoting the sedative effect of midazolam in the neonatal rats was largely KCC2-dependent.”

　　after “There was a significant interaction between the effect of CLP290 and VU0463271 (p = 0.011). The main effect of CLP290 was significant (p < 0.001), whereas that of VU0463271 was not significant (p = 0.785). A post-hoc median regression analysis was performed as follows: First, the righting reflex latency after midazolam injection with CLP290 pretreatment was significantly longer than that without CLP290 administration (Fig 5B). Furthermore, VU0463271 coadministration significantly attenuated the effect of CLP290 on the righting reflex (Fig 5B). These results suggest that the effect of CLP290 on midazolam-induced sedation in neonatal rats is largely KCC2-dependent.”

---

## [Decision Letter · Decision Letter 1]

22 Feb 2021

CLP290 promotes the sedative effects of midazolam in neonatal rats in a KCC2-dependent manner: A laboratory study in rats

PONE-D-20-09548R1

Dear Dr. Miyazaki,

We’re pleased to inform you that your manuscript has been judged scientifically suitable for publication and will be formally accepted for publication once it meets all outstanding technical requirements.

Kind regards,

Giuseppe Biagini, MD

Academic Editor

PLOS ONE

Additional Editor Comments (optional):

Reviewers' comments:

Reviewer's Responses to Questions

**Comments to the Author**

1. If the authors have adequately addressed your comments raised in a previous round of review and you feel that this manuscript is now acceptable for publication, you may indicate that here to bypass the “Comments to the Author” section, enter your conflict of interest statement in the “Confidential to Editor” section, and submit your "Accept" recommendation.

Reviewer #1: All comments have been addressed

Reviewer #2: All comments have been addressed

2. Is the manuscript technically sound, and do the data support the conclusions?

Reviewer #1: (No Response)

Reviewer #2: Yes

3. Has the statistical analysis been performed appropriately and rigorously? 

Reviewer #1: (No Response)

Reviewer #2: I Don't Know

4. Have the authors made all data underlying the findings in their manuscript fully available?

Reviewer #1: (No Response)

Reviewer #2: Yes

5. Is the manuscript presented in an intelligible fashion and written in standard English?

Reviewer #1: (No Response)

Reviewer #2: Yes

6. Review Comments to the Author

Reviewer #1: (No Response)

Reviewer #2: (No Response)

7. PLOS authors have the option to publish the peer review history of their article (what does this mean?). If published, this will include your full peer review and any attached files.

Reviewer #1: No

Reviewer #2: No

---

## [Editor Report · Acceptance letter]

3 Mar 2021

PONE-D-20-09548R1 

CLP290 promotes the sedative effects of midazolam in neonatal rats in a KCC2-dependent manner: A laboratory study in rats 

Dear Dr. Miyazaki:

I'm pleased to inform you that your manuscript has been deemed suitable for publication in PLOS ONE. Congratulations! Your manuscript is now with our production department. 

Kind regards, 

on behalf of

Dr. Giuseppe Biagini 

Academic Editor

PLOS ONE